# Effectiveness and Safety of Fufang Danshen Dripping Pill (Cardiotonic Pill) on Blood Viscosity and Hemorheological Factors for Cardiovascular Event Prevention in Patients with Type 2 Diabetes Mellitus: Systematic Review and Meta-Analysis

**DOI:** 10.3390/medicina59101730

**Published:** 2023-09-27

**Authors:** Minji Wi, Yumin Kim, Cheol-Hyun Kim, Sangkwan Lee, Gi-Sang Bae, Jungtae Leem, Hongmin Chu

**Affiliations:** 1College of Korean Medicine, Wonkwang University, Iksandaero 460, Iksan 54538, Jeollabuk-do, Republic of Korea; minlove6245@naver.com (M.W.); tjssu682@gmail.com (Y.K.); 2Department of Internal Medicine and Neuroscience, College of Korean Medicine, Wonkwang University, Iksandaero 460, Iksan 54538, Jeollabuk-do, Republic of Korea; user2307@hanmail.net (C.-H.K.); sklee@wku.ac.kr (S.L.); 3Department of Pharmacology, School of Korean Medicine, Wonkwang University, Iksandaero 460, Iksan 54538, Jeollabuk-do, Republic of Korea; baegs888@wku.ac.kr; 4Hanbang Cardio-Renal Syndrome Research Center, School of Korean Medicine, Wonkwang University, Iksandaero 460, Iksan 54538, Jeollabuk-do, Republic of Korea; 5Korean Traditional Medicine Institute, Wonkwang University, Iksandaero 460, Iksan 54538, Jeollabuk-do, Republic of Korea; 6Research Center of Traditional Korean Medicine, College of Korean Medicine, Wonkwang University, Iksandaero 460, Sin-dong, Iksan 54538, Jeollabuk-do, Republic of Korea; 7Wollong Public Health Subcenter, Paju Public Health Center, Paju 10924, Gyeonggi-do, Republic of Korea

**Keywords:** systematic review, meta-analysis, type 2 diabetes mellitus, Fufang danshen dripping pill, blood viscosity

## Abstract

*Background and Objectives*: Diabetes can cause various vascular complications. The Compounded Danshen-Dripping-Pill (CDDP) is widely used in China. This study aimed to analyze the effectiveness and safety of CDDP in the blood viscosity (BV) with type 2 diabetes mellitus (T2DM). *Materials and Methods*: We conducted a systematic search of seven databases from their inception to July 2022 for randomized controlled trials that used CDDP to treat T2DM. To evaluate BV, we measured low shear rate (LSR), high shear rate (HSR), and plasma viscosity (PV). Homocysteine and adiponectin levels were also assessed as factors that could affect BV. *Results*: We included 18 studies and 1532 patients with T2DM. Meta-analysis revealed that CDDP significantly reduced LSR (mean difference [MD] −2.74, 95% confidence interval [CI] −3.77 to −1.72), HSR (MD −0.86, 95% CI −1.08 to −0.63), and PV (MD −0.37, 95% CI −0.54 to −0.19) compared to controls. CDDP also reduced homocysteine (MD −8.32, 95% CI −9.05 to −7.58), and increased plasma adiponectin (MD 2.72, 95% CI 2.13 to 3.32). Adverse events were reported less frequently in the treatment groups than in controls. *Conclusions*: CDDP is effective in reducing BV on T2DM. However, due to the poor design and quality of the included studies, high-quality, well-designed studies are required in the future.

## 1. Introduction

Diabetes mellitus is a metabolic disease characterized by insulin resistance in the target organs or an absolute or relative insulin deficiency and can be classified into type 1 diabetes mellitus, type 2 diabetes mellitus (T2DM), and gestational diabetes [1]. Diabetes is a major cause of numerous micro- and macrovascular complications that not only reduce the quality of life and life expectancy, but can also lead to blindness, kidney failure, myocardial infarction, stroke, and limb amputation [2]. The prevalence of diabetes worldwide is increasing, and the International Diabetes Federation (IDF) estimates that the number of people with diabetes will reach 783.2 million in 2045 and the number of people with diabetes will increase by 46% by 2045, which is expected to reach 783.2 million people [3,4,5]. Currently, the global prevalence is estimated to be over 10.5%, and the increase in prevalence is particularly rapid in Asia. The increase in the diabetic population is associated with changes in lifestyle factors, including increased prevalence of overweight and obesity, social factors such as smoking, westernized diets, and changes to sedentary lifestyles [6]. As the number of patients with diabetes increases, there is increasing pressure on the national health system to treat diabetes and diabetic complications; therefore, prevention and early treatment of diabetes are important. As the prevalence of T2DM increases, so does the risk of cardiovascular disease [7,8,9]. Adults with diabetes are known to have a 2–4 times increased cardiovascular risk compared with adults without diabetes [10]. The ultimate goal of diabetes mellitus management is to prevent various complications and lower the mortality rate. Conventional therapies aiming for this primarily include the drugs Metformin, DDP-4 inhibitor, SGLT2-inhibitor, Sulfonylureas, and Thiazolidinediones. If blood sugar levels are not controlled with these oral medications, insulin injection is sometimes used [11,12]. However, these conventional medicines have a number of side effects: there have been cases of hypoglycemia with the use of sulfonylurea, there is a report that an alpha-glucosidase inhibitor causes liver dysfunction, and acute pancreatitis or joint dysfunction has been reported with the use of DDP-4 [13,14]. Therefore, a complementary therapeutic option is needed for use with existing hyperglycemic agents when blood sugar levels are controlled poorly or to reduce the occurrence of diabetic complications [15]. 

Oxidative stress plays an important role in cardiovascular disease in diabetic patients [16]. Among blood rheological properties, blood viscosity is known to be a major predictor of oxidative stress, previous studies have shown that blood viscosity is elevated in people with high oxidative stress, such as lead-exposed laborers or smokers, compared to healthy individuals [17,18]. In addition, it is known that blood viscosity is higher in patients with type 2 diabetes than in healthy people [19,20]. Therefore, effective management of blood viscosity is required to prevent cardiovascular disease in patients with T2DM; however, no drugs are known to be effective against blood viscosity. 

In East Asia, various herbal medicines and acupuncture are being used to treat diabetes and its associated complications [21,22]. Among these herbal prescriptions, the Compounded Danshen Dripping Pill (CDDP; also known as the cardiotonic pill) is a drug used for coronary vascular disease management in China and has been reported to be effective in lowering blood sugar and treating diabetic complications, including blood viscosity (Figure 1). In particular, CDDP has been shown to suppress oxidative stress and inflammatory responses in animal experiments [23,24]. Clinical studies have also reported that CDDP is effective in reducing triglyceride or low-density lipoprotein cholesterol (LDL-C) levels and slowing the progression of diabetic retinopathy when used in combination with aspirin in a coronary disease case [25,26]. However, there is no comprehensive review on the effect of CDDP on blood oxidative stress in diabetic patients, and there is no study on the effect of CDDP on blood viscosity and related hemorheological indices. Therefore, this study aimed to analyze the effectiveness and safety of CDDP in T2DM via a systematic review to summarize the evidence and suggest implications for further study and clinical practice.

## 2. Materials and Methods

### 2.1. Study Registration

The systematic literature review protocol was prepared according to the Preferred Reporting Items for Systematic Review and Meta-Analysis Protocols (PRISMA-P) guidelines. The systematic review protocol was registered in the International Prospective Register of Systematic Reviews (PROSPERO; Registration ID: CRD42022352381). Since this study quantitatively synthesized data from previously published papers, institutional review board approval and participant consent were not required.

#### Searching Strategy

The following seven databases were searched from their inception to September 2022: MEDLINE (PubMed, https://pubmed.ncbi.nlm.nih.gov/ accessed on 30 September 2022), Cochrane Library (CENTRAL, https://www.cochranelibrary.com/), EMBASE (https://www.embase.com), OASIS (https://oasis.kiom.re.kr/), Korea Citation Index (KCI; https://www.kci.go.kr), and China National Knowledge Infrastructure (CNKI; https://oversea.cnki.net/index/), Research Information Sharing Service (RISS; http://www.riss.kr/index.do). The search strategies are shown in Appendix A.

### 2.2. Eligibility Criteria for Study Selection

#### 2.2.1. Types of Studies

The systematic review and meta-analysis included published peer-reviewed randomized controlled trials (RCTs). We manually searched for studies of patients with type 2 diabetes who were treated with CDDP and checked blood viscosity for potential inclusion.

#### 2.2.2. Types of Participants

Eligible participants were patients diagnosed with T2DM. There were no restrictions based on sex, ethnicity, symptom severity, disease duration, or clinical environment. 

#### 2.2.3. Types of Interventions and Comparators

The research question of this review was ‘Does CDDP have a significant effect on blood viscosity, homocysteine, and plasma adiponectin levels in T2DM patients?’ The definition of CDDP was limited to the ‘Fufang danshen dripping pill’ of Tasly Pharm Co., Tianjin, China. Other names for CDDP, such as ‘cardiotonic pills’ and ‘Fufang danshen dripping pill,’ were included in the search strategy. The control group was defined as conventional Western medicine therapy for diabetes and included studies that used a placebo group administered a herbal medicine that had the same appearance and flavor as CDDP. The included studies were limited to oral medications, excluding other routes of administration such as injections.

#### 2.2.4. Types of Outcome Measures

##### Primary Treatment Outcome: Final Value of Blood Viscosity (Low and High Shear Rate) 

For the confirmation of blood viscosity, the cone plate rotation and scanning capillary methods were considered representative. The cone plate rotation method is a technique that measures the blood resistance when a certain amount of blood is put between the cone and the plate and rotated, and converts it into blood viscosity [27]. In the case of the scanning capillary method, the blood is passed through an Ethylenediaminetetraacetic acid (EDTA) tube and the viscosity of the blood is measured. In the cone plate rotation method, high blood viscosity (known as systolic blood viscosity or high shear rate; HSR) (300 s^−1^) was 3.25–4.91 for men and 2.94–4.59 for women, and low blood viscosity (known as diastolic blood viscosity or low shear rate; LSR) (5 s^−1^) was 7.75–11.48 for men and 7.23–10.61 for women [28]. In the scanning capillary method, the HSR was 3.5–4.1 in men and 3.0–3.6 in women, and the LSR was 9.35–13.1 in men and 7.59–11.13 in women [29]. Factors affecting blood viscosity include not only blood glucose, but also fibrinogen, red blood cells (RBCs), white blood cells (WBCs), and triglyceride levels [17,30,31]. A recent study showed that blood viscosity at a low shear rate was associated with the occurrence of early neurological deterioration in patients with lacunar infarction [32]. Furthermore, hyper-viscosity can occur with COVID-19 infection, causing poor tissue perfusion, peripheral vascular resistance, and thrombosis [33]. 

##### Secondary Treatment Outcomes: Plasma Viscosity, Homocysteine, and Plasma Adiponectin Levels

Plasma viscosity: The quantity of proteins in the blood has an impact on the viscosity of plasma [34]. The normal plasma viscosity range is 1.10–1.30 mPas at 37.7 °C [35]. 

Homocysteine: Homocysteine is a sulfur-containing amino acid and is generated from the breakdown of the dietary amino acid methionine [36]. Excessive accumulation of homocysteine can increase the risk of brain infarction or dementia and is a risk factor for atherosclerosis [37,38]. Furthermore, higher homocysteine levels were observed in T2DM patients than in controls, indicating that a higher homocysteine level is a novel risk factor for predicting diabetic complications such as diabetic retinopathy [39]. Since homocysteine is a type of amino acid associated with hyper-viscosity, along with multiple other factors related to blood viscosity, it was investigated as a secondary outcome [40].

Adiponectin: Adiponectin, also known as adipocyte complement-related protein of 30 kDa (Acrp30) or AdipoQ, is a 244-amino acid protein secreted mainly by adipose tissue. Adiponectin is a hormone that initiates the use of body fat for energy, which is found primarily in white fat tissue and less in brown fat tissue [41]. People with higher levels of adiponectin are less likely to develop cardiovascular disease than those with low levels [42]. Moreover, some studies reported that higher levels of adiponectin are associated with a lower risk of high blood pressure, cardiovascular disease, and obesity [43,44,45]. Circulating adiponectin levels have been correlated with factors affecting blood viscosity, such as RBC deformability and systemic inflammatory markers [46,47]. Thus, this study investigated adiponectin as a secondary outcome. 

##### Primary Safety Outcome: Rate of Adverse Events

To measure the adverse events rate, the study verified adverse events reported in the included clinical trials and compared the adverse events rate between the control group and the CDDP group.

#### 2.2.5. Data Extraction and Risk of Bias Assessment

Two independent authors (MY and YK) performed data extraction and quality assessment of the RCTs using a data extraction form and Excel software (version 2201). The form included the year of study publication, participant characteristics, sample size, duration of treatment, frequency of treatment, type of comparator, type of outcomes, and adverse events. 

The risk of bias was assessed using the tool from the Cochrane Handbook Version 6.0, which included random sequence generation, allocation concealment, blinding of the participants and personnel, blinding of the outcome assessments, incomplete outcome data, selective reporting, and other sources of bias [48]. Random sequence generation was evaluated as ‘low risk’ when the random sequence of the study was described in detail, and as ‘unclear risk’ if no other specific methodology was mentioned. In allocation concealment, studies in which it was unknown whether a third researcher was assigned were evaluated as ‘unclear risk’. In blinding of participants and personnel, studies prescribing CDDP as an add-on medicine in the treatment group were evaluated as ‘high risk’ because the patient and researcher could have known which group the subject belonged to, and studies using placebo were evaluated as ‘low risk’. Blinding outcome assessment was generally evaluated as ‘low risk’ because it was an objective blood test result, such as blood viscosity, homocysteine, and plasma adiponectin. In incomplete outcome data, studies with no missing values or that indicated the reason for dropout were evaluated as ‘low risk’, and studies that did not indicate the reason for the patient’s dropout were evaluated as ‘unclear risk’. In selective reporting, publications where the study protocol could not be found were evaluated as ‘unclear risk’; all included studies were evaluated as ‘unclear risk’ because the protocol was not searched. In other sources of bias, studies were considered ‘low risk’ when there were no factors that could affect the research results, such as conflicting results with research from other sources (following research quality evaluation) or when the research was sponsored by a specific company. Two authors independently verified the outcomes of this procedure and any discrepancies were resolved through discussion. 

#### 2.2.6. Data Synthesis

For the quantitative synthesis of continuous variables, we calculated the mean difference (MD) and 95% confidence interval (CI) for the final values. Heterogeneity was assessed using the I2 statistic, and a random-effects model was used to account for any observed heterogeneity. Publication bias was tested using funnel plots and Egger’s test [49]. 

We assessed the overall quality of the evidence using the GRADE approach. The quality of evidence was rated as high, moderate, low, or very low based on the study design, risk of bias, indirectness, imprecision, inconsistency, and publication bias. Generally, we rated the quality of evidence as moderate owing to some heterogeneity observed in the data synthesis and the possibility of publication bias [50].

## 3. Results

### 3.1. Study Selection

A total of 2649 articles were initially identified from six electronic databases. After excluding duplications, irrelevant studies, and review articles, 173 potentially eligible articles were selected. Finally, 18 studies including 1532 patients with T2DM were included (Figure 2 and Appendix A).

### 3.2. Study Characteristics 

The included studies were conducted in China and were published between 2005 and 2020. All studies were conducted on patients with diabetes. In particular, when classified by disease including duplicates, eight studies included patients with myocardial ischemia as a diabetic complication [51,52,53,54,55,56,57,58], one with diabetic retinopathy (with nephropathy) [59], six with diabetic nephropathy [59,60,61,62,63,64], and two with diabetic neuropathy [65,66]. Others were related to type 2 diabetes (Appendix A).

According to the intervention method, when the included clinical trials were classified, ten studies compared the effects between conventional Western medicine therapies and CDDP (head-to-head design) [51,53,54,55,56,57,58,65,66,67], and seven studies compared the effects between combined CDDP and Western medicine therapy and single Western medicine therapy [52,59,61,62,63,64,68]. Among them, two studies included dietary control in the treatment and control groups [59,60]. The final study compared the effects between combined CDDP and Western medicine therapy and combined placebo CDDP and Western medicine therapy [60].

### 3.3. Quality of Evidence and Publication Bias

To assess the risk of bias in our meta-analysis, we evaluated the included studies based on key criteria, including random sequence generation, allocation concealment, blinding of participants, blinding of outcome assessment, incomplete outcome data, and selective reporting. Our assessment revealed that 2/18 studies had a low risk of bias for random sequence generation [53,66]. The high number of unclear risk studies was due to the lack of clear information provided regarding the methods used for randomization. For allocation concealment, only one study clearly reported the method used and was assessed as having a low risk of bias [64]. With regards to the blinding of participants, only one study was assessed as having a low risk of bias, while the rest were assessed as having a high risk [60]. This was because the intervention and control groups received different interventions, making it impossible to blind the participants. Blinding of the outcome assessment was evaluated as having a low risk of bias for all studies included in our analysis. As the outcome of interest was a blood test, it was impossible for the participants or researchers to influence the outcome. In incomplete outcome data assessment, we found four studies had an unclear risk of bias for incomplete outcome data [51,54,64,68]. Finally, selective reporting was assessed as having a low risk of bias for all studies included in our analysis. This was because the number of participants in the intervention and control groups was reported before and after the intervention, ensuring that there was no selective reporting (Figure 3 and Figure 4).

### 3.4. Effectiveness of CDDP on Blood Viscosity

Studies were classified into subgroups according to the study design: (1) CDDP + Western Medicine (WM) (treatment group) vs. WM alone (control group), (2) CDDP alone (treatment group) vs. WM alone (control group), and (3) CDDP + WM (treatment group) vs. Placebo CDDP + WM (control group). For evaluating the effectiveness of CDDP on blood viscosity, five variables were measured: LSR, HSR, and plasma viscosity were measured as primary outcomes and homocysteine and adiponectin were measured as related factors. Publication bias according to individual variables is shown in Appendix A. The results of the GRADE assessment are shown in Appendix A.

#### 3.4.1. Low Shear Rate

When compared with controls, LSR was significantly lower in the CDDP group (7 studies, MD −2.74, 95% CI −3.77 to −1.72) (Figure 5A) [59,60,61,64,66,67,68]. LSR was then analyzed in the individual subgroups according to the study design. In the four studies using CDDP + WM vs. WM, LSR was significantly lower in the CDDP + WM group than in the WM group (MD −2.97, 95% CI −3.76 to −2.19) [59,61,64,68]. In the two studies using CDDP vs. WM, LSR was also significantly lower in the CDDP group than in the WM group (MD −3.17, 95% CI −3.91 to −3.50) [66,67]. In the one study using CDDP + WM vs. Placebo CDDP + WM, there was no significant difference between the groups (MD −0.28, 95% CI −0.98 to 0.42) [60].

#### 3.4.2. High Shear Rate

When compared with controls, HSR was significantly lower in the CDDP group (7 studies, MD −0.86, 95% CI −1.08 to −0.63) (Figure 5B) [59,60,61,64,66,67,68]. In subgroup analysis, four studies showed HSR was significantly lower in the CDDP + WM treatment group than in the WM control group (MD −1.03, 95% CI −1.16 to −0.90) [59,61,64,68]. In the two studies using CDDP vs. WM, HSR was significantly lower in the treatment group than in the control group (MD −0.84, 95% CI −1.09 to −0.59) [66,67]. In the one study comparing CDDP + WM vs. Placebo CDDP + WM, HSR was significantly lower in the treatment group than in the control group (MD −0.20, 95% CI −0.54 to 0.14) [60].

#### 3.4.3. Plasma Viscosity

When the results of 10 studies measuring plasma viscosity (PV) were compared, the PV was lower in the treatment group than in the control group (MD −0.37, 95% CI −0.54 to −0.19) (Figure 5C) [59,60,61,62,63,64,65,66,67,68]. In subgroup analysis, six studies showed PV was significantly lower in the CDDP + WM treatment group than in the WM control group (MD −0.39, 95% CI −0.69 to −0.10) [59,61,62,63,64,68]. In the three studies using CDDP vs. WM, PV was significantly lower in the treatment group than in the control group (MD −0.34, 95% CI −0.61 to −0.08) [65,66,67]. In the one study using CDDP + WM vs. Placebo CDDP + WM, PV was significantly lower in the treatment group than in the control group (MD −0.22, 95% CI −0.57 to 0.13) [60].

### 3.5. Homocysteine

A meta-analysis of the eight studies measuring homocysteine showed lower homocysteine levels in the treatment group than in the control group (MD −8.32, 95% CI −9.05 to −7.58) (Figure 6) [51,52,53,54,55,56,57,58]. In subgroup analysis, homocysteine was significantly lower in the CDDP + WM treatment group than in the WM control group (n = 6, MD −8.23, 95% CI −9.08 to −7.38) [53,54,55,56,57,58]. Similarly, homocysteine was significantly lower in the CDDP treatment group than in the WM control group (n = 2, MD −8.55, 95% CI −9.99 to −7.12) [51,52]. No studies in the CDDP + WM vs. Placebo CDDP + WM subgroup reported homocysteine levels.

### 3.6. Plasma Adiponectin Levels

Eight studies measured plasma adiponectin levels. Among them, the study by Lu was excluded from the meta-analysis because the results suggested that there was an error in the notation of research data values [55]. When the analysis was conducted on the remaining seven studies, plasma adiponectin levels were higher in the treatment group than in the control group (MD 2.72, 95% CI 2.13 to 3.32) (Figure 7) [51,52,53,54,56,57,58]. In subgroup analysis, plasma adiponectin levels were significantly higher in the CDDP + WM treatment group than in the WM control group (n = 5, MD 2.60, 95% CI 1.87 to 3.34) [53,54,56,57,58]. Similarly, plasma adiponectin levels were significantly higher in the CDDP group than in the WM control group (n = 2, MD 3.03, 95% CI 1.57 to 4.49) [51,52]. No studies in the CDDP + WM vs. Placebo CDDP + WM subgroup reported plasma adiponectin levels.

### 3.7. Safety

There were 6/18 studies that measured adverse events (Appendix A) [53,59,62,64,67,68]. One study reported no adverse events in both the treatment group and the control group [64]. In the remaining six studies, headache (n = 1), nausea (n = 1), dizziness (n = 1), and diarrhea (n = 1) were reported as adverse events in the treatment group. Overall, there were fewer adverse events in the treatment group than in the control group.

## 4. Discussion

### 4.1. Summary of Findings

In this systematic review and meta-analysis, the effectiveness of CDDP on the blood viscosity of patients with type 2 diabetes was searched in six databases. A total of 18 studies including 1532 DM patients were selected and investigated. The main findings of this review were that, compared to the WM control group, CDDP or CDDP + WM treatment in patients with T2DM resulted in significantly lower LSR, HSR, PV, and hemorheologic factors, such as homocysteine and adiponectin.

### 4.2. Importance of Blood Viscosity Control in DM Management

Diabetes mellitus is a type of metabolic disease in which the secretion or normal function of insulin is insufficient, and is characterized by an increase in the concentration of glucose in the blood, called hyperglycemia [69]. Chronic hyperglycemia damages blood vessels and disrupts the formation of micro-vessels. For this reason, kidney or cardiovascular problems are common diabetic complications [70]. Diabetes is a chronic disease that is difficult to prevent, making management a social burden [5,71]. Therefore, the prevention and early management of diabetes is important [69]. Recently, studies have shown that elevated blood viscosity is frequently found in patients with diabetes, and they are among the causes of diabetic complications. Therefore, managing blood viscosity is helpful in preventing the progression or worsening of diabetes [19,72,73].

### 4.3. Blood Viscosity, a Concept Similar to Blood Stasis, Is a Main Pathophysiology of East Asia Traditional Medicine

In East Asian medicine, various attempts have been made to improve the condition of the blood. High blood viscosity, called “blood stasis” in traditional East Asian medicine, was thought to be one of the leading causes of chest pain and cerebral infarction and is listed in the International Classification of Diseases (ICD) code as “blood stasis pattern” or “blood stagnation pattern” [74,75]. This blood stasis pattern is derived from the concept of high blood viscosity and is believed to share similarities with the causes of metabolic and chronic pain diseases [76,77]. In particular, various drugs have been developed to improve blood conditions, including blood viscosity. CDDP, the target of this study, is a drug for reducing cardiovascular diseases, including hyperlipidemia and hypertension, and improves factors that affect blood viscosity [23,24,26]. In fact, CDDP is widely used to treat diabetic complications, diabetic retinopathy, hyperlipidemia, and coronary artery pathway conditions [26]. The main mechanisms of CDDP are microcirculatory recovery, anti-oxidative, anti-inflammatory, and inhibition of platelet adhesion and aggregation [24,78]. CDDP has completed Phase II clinical trials in the United States of America and China. Blood viscosity management is important; however, there is a lack of conventional medications with sufficient clinical evidence to support their effectiveness in treating blood viscosity [79]. Therefore, several alternative therapeutic options, such as exercise, diet, or yoga, are recommended to attempt to control blood viscosity [80,81,82].

### 4.4. Predicted Mechanism Affecting Blood Viscosity-Related Factors in CDDP

Various interventions have been utilized to regulate blood viscosity for disease prevention and treatment. Several reports have shown that natural products lower blood viscosity, and Galduróz et al. confirmed a decrease in blood viscosity after using Ginkgo biloba for 180 days [83]. In addition, animal experiments have demonstrated that *Salvia miltiorrhiza* extract lowers blood viscosity [84,85]. CDDP consists of 9 mg *S. miltiorrhiza*, along with 1.76 mg Panax notoginseng and Borneolum. Further, its major active constituents are tanshinol, protocatechuic aldehyde, salvianolic acid B, and notoginsenoside [86]. A previous study demonstrated a dose-dependent inhibition of Adenosine diphosphate(ADP)-induced platelet aggregation in vivo by salvianolic acid, a component of *S. miltiorrhiza* in CDDP. In addition, tanshinol shows similar results as aspirin in inhibiting Cox-2 and exerts its effect through the down-regulation of thromboxane B2 [87]. Unlike Aspirin, tanshinol has a protective effect against ulcer formation, which is advantageous over conventional aspirin therapy. As such, CDDP is expected to improve blood viscosity by reducing oxidative stress, increasing microcirculation, protecting blood vessels, and anti-inflammatory and anti-apoptotic effects [23,78]. Research on natural-based candidate substances for the treatment of diabetes complications is also conducted in silico, so future in-silico-based studies on the constituents of CDDP may aid in understanding their mechanisms [88]. In particular, as confirmed in this study, CDDP reduces the levels of homocysteine and plasma adiponectin. This may result from a multi-component, multi-target effect on the metabolism of these blood proteins, which is presumed to reduce blood viscosity and increase microcirculation. The summary of the mechanism of action of CDDP is depicted in Figure 8.

### 4.5. Strengths

This study had several limitations. Firstly, this study set blood viscosity as the primary outcome and did not verify the difference between hematocrit and fibrinogen, which are also important factors influencing blood viscosity. Because there are numerous factors that affect blood viscosity, it is necessary to conduct a meta-analysis on all factors affecting blood viscosity in the future. Moreover, blood viscosity itself has limitations; the normal range is still quite wide according to the current test method and the value is affected by various factors. Secondly, several included studies had various methodological flaws; therefore, there is a need for high-quality, well-designed studies in the future. Additionally, the information presented in some articles was inaccurate. For example, the study by Lu presented the SD of homocysteine as 11.25, which was higher than the average value of 9 [55]; it was hypothesized that the publication incorrectly indicated 1.25 as 11.25. However, we could not directly modify the RCT values of Lu while conducting the meta-analysis, so this study was excluded [55]. Thirdly, as this study targeted only type 2 diabetic patients, additional research is needed to determine if CDDP improves blood viscosity in diabetic patients with genetic predispositions, such as type 1 diabetes. Furthermore, we searched several core databases, including Pubmed, EMBASE, and CENTRAL, but we did not review all databases such as Web of Science. Therefore, it is possible that studies not included in our search may exist.

### 4.6. Limitations

There are also several strengths of our study. Although CDDP is widely used for the treatment of underlying diseases that affect blood viscosity, there has been no comprehensive review of the effectiveness of CDDP on blood viscosity. Therefore, this is the first study to identify studies in which blood viscosity was measured and assess the effectiveness and safety of CDDP in patients with diabetes by measuring blood viscosity. In addition, homocysteine and adiponectin, which are known to be related to blood viscosity, were evaluated as secondary outcomes, supporting the research results. In the case of diabetic patients included in the sample, it was observed that blood viscosity was higher in both systolic and diastolic periods than the normal reference value, and blood viscosity decreased after taking CDDP. Moreover, the homocysteine level was higher than the average value of 5–15 mmol/L, and the homocysteine level was significantly lower in the CDDP treatment group. These results are useful for supporting physicians in making clinical decisions.

### 4.7. Implications for Clinical Practice and Future Research

Homocysteine and plasma adiponectin levels, as well as values related to blood viscosity, are all affected by sex differences; however, these issues were not considered in this study. In future clinical studies that use CDDP as an intervention and blood viscosity as the outcome, we recommend conducting a pilot study with healthy men aged 20–30 and excluding women whose blood test values may be affected by hormones produced during the menstrual cycle. More research is required to derive the normal range values for blood viscosity, especially classifying groups according to variables that may affect blood viscosity, including race, sex, age and obesity [89]. In addition, since most studies evaluated surrogate outcomes, it is necessary to plan RCTs that apply long-term follow-up clinical endpoints, such as diabetic complications or death. In terms of clinical practice, we believe that blood viscosity will become an increasingly important area in the future; blood viscosity values could be used to predict the occurrence risk of cardiovascular and cerebrovascular diseases. Our study suggests the possibility of CDDP as a candidate drug for blood viscosity management. Early prevention and management of complications in patients with diabetes is very important; therefore, prevention of vascular complications by concurrently administering CDDP as early as possible in patients with high blood viscosity may help to limit further complications from diabetes.

## 5. Conclusions

Our meta-analysis revealed the treatment with CDDP decreased blood viscosity in diabetic patients and reduced homocysteine and plasma adiponectin levels, which are factors that can affect blood viscosity, without severe adverse events. Therefore, CDDP combined treatment could be an effective and safe therapeutic method for controlling blood viscosity in patients with type 2 diabetes. However, as the methodological quality was relatively low, further well-designed, long-term follow-up clinical trials are required to improve confidence in this conclusion.

## Figures and Tables

**Figure 1 medicina-59-01730-f001:**
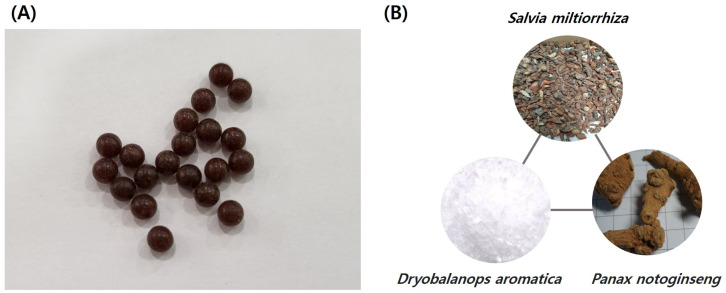
Compounded Danshen Dripping Pills (CDDPs). (**A**) CDDPs are a brown colored, round pill, usually 4.5 mm (0.18 inch) in size. (**B**) The three main herbal components of CDDP are *Salvia miltiorrhiza*, *Dryobalanops aromatica*, and *Panax notoginseng*.

**Figure 2 medicina-59-01730-f002:**
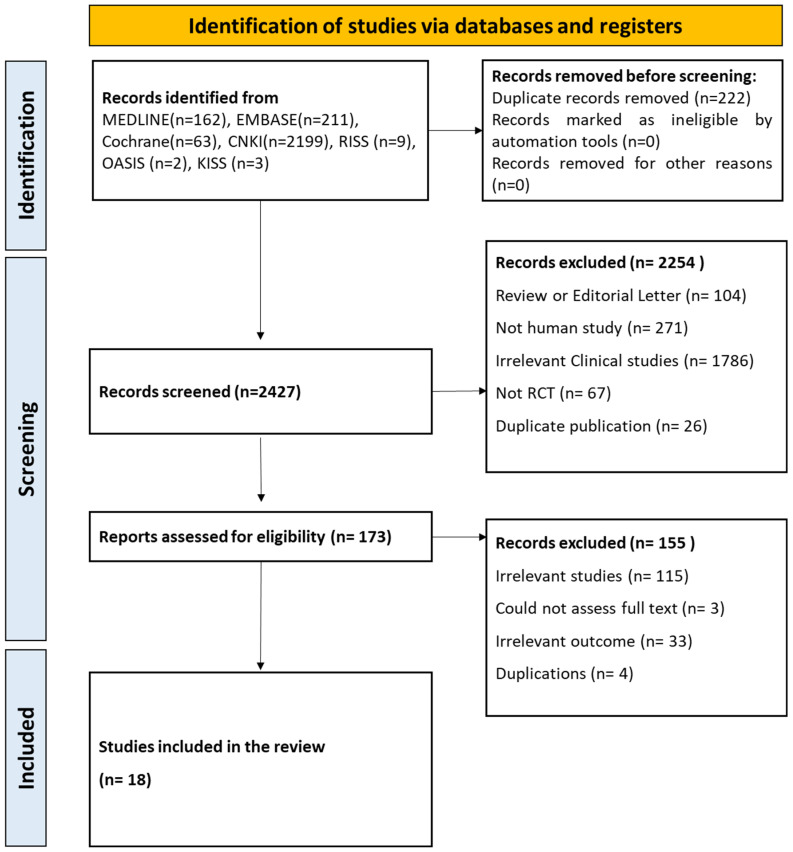
Flowchart of this study.

**Figure 3 medicina-59-01730-f003:**
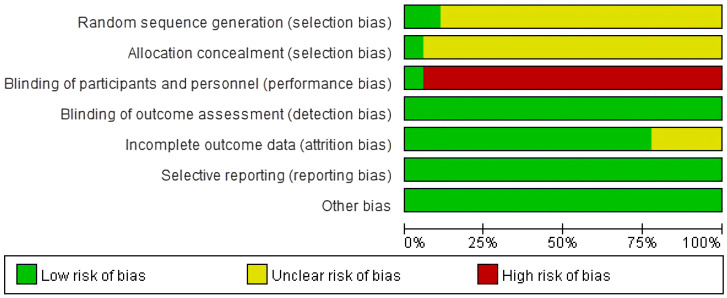
Risk of bias assessment.

**Figure 4 medicina-59-01730-f004:**
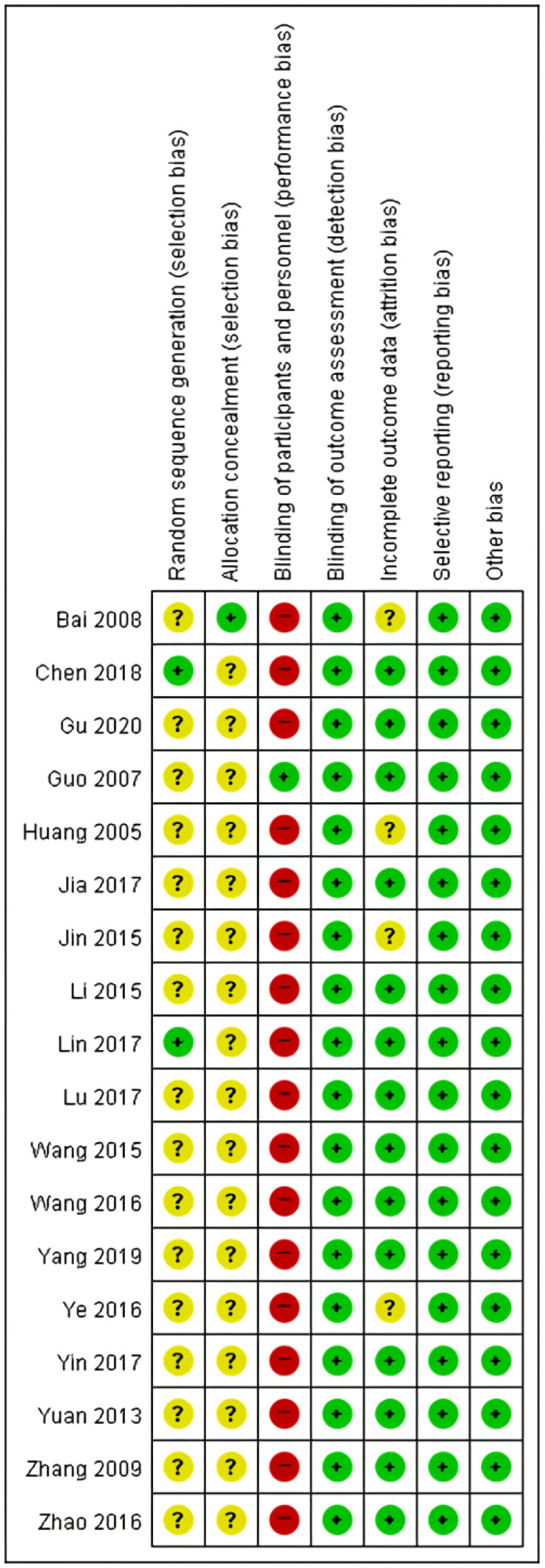
Risk of bias summary for the 18 studies analyzed. Green ‘+’ circles = Low risk of Bias; yellow ‘?’ circles = uncertainties about the risk of bias; red ‘–’ circles = high risk of bias. Reference: Bai 2008 [64]; Chen 2018 [53]; Gu 2020 [65]; Guo 2007 [60]; Huang 2005 [68]; Jia 2017 [63]; Jin 2015 [54]; Li 2015 [56]; Lin 2017 [66]; Lu 2017 [55]; Wang 2015 [58]; Wang 2016 [57]; Yang 2019 [62]; Ye 2016 [51]; Yin 2017 [67]; Yuan 2013 [61]; Zhang 2009 [59]; Zhao 2016 [52].

**Figure 5 medicina-59-01730-f005:**
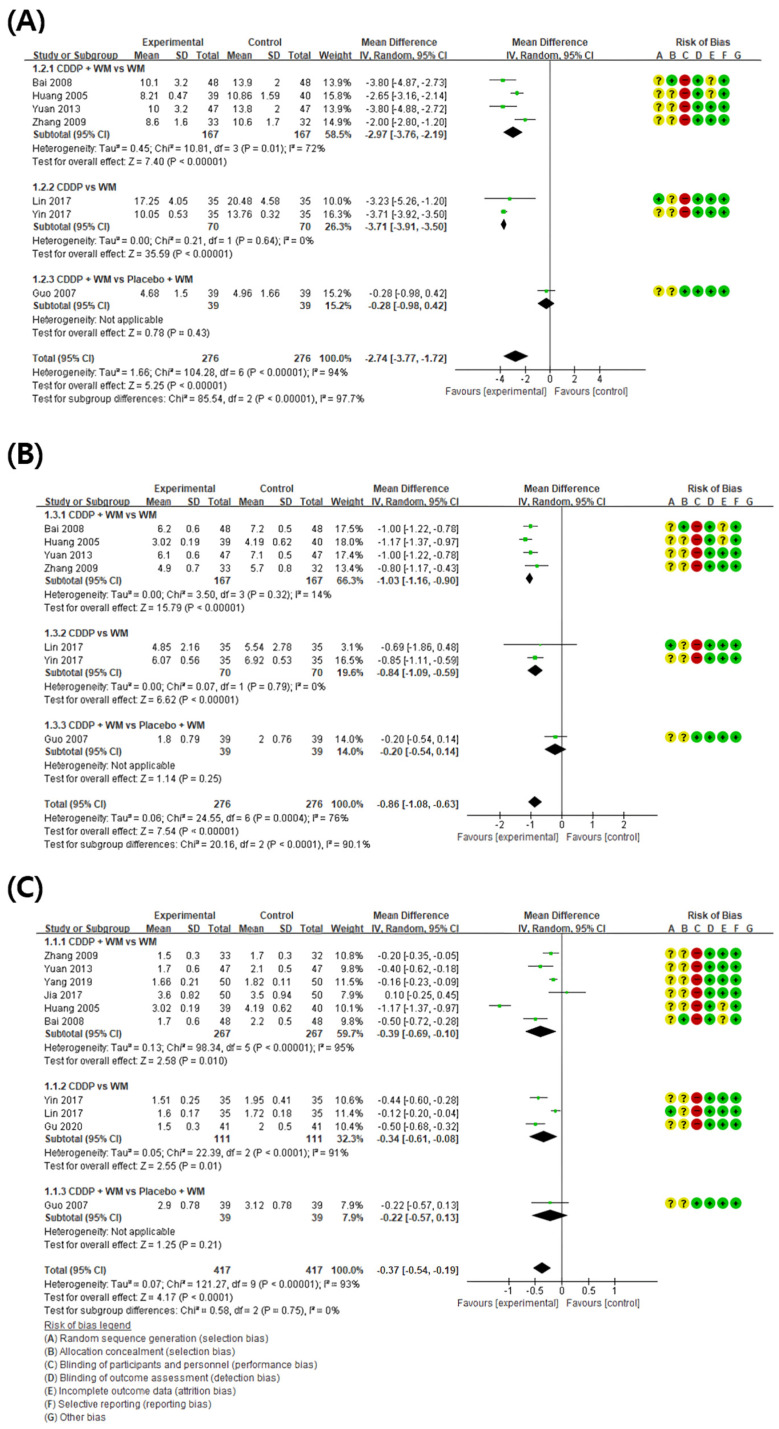
Meta-analysis of blood viscosity factors between treatment groups and control groups. (**A**) Low shear rate, (**B**) high shear rate, and (**C**) plasma viscosity. Green ‘+’ circles = Low risk of Bias; yellow ‘?’ circles = uncertainties about the risk of bias; red ‘−’ circles = high risk of bias. The black rectangle represents the mean difference; The green dot and bar represent the overall summary estimate of the treatment effect, especially, the bar represents the confidence interval for the summary estimate. Reference: Bai 2008 [64]; Huang 2005 [68]; Yuan 2013 [61]; Zhang 2009 [59]; Lin 2017 [66]; Yin 2017 [67]; Guo 2007 [60]; Yang 2019 [62]; Jia 2017 [63]; Gu 2020 [65].

**Figure 6 medicina-59-01730-f006:**
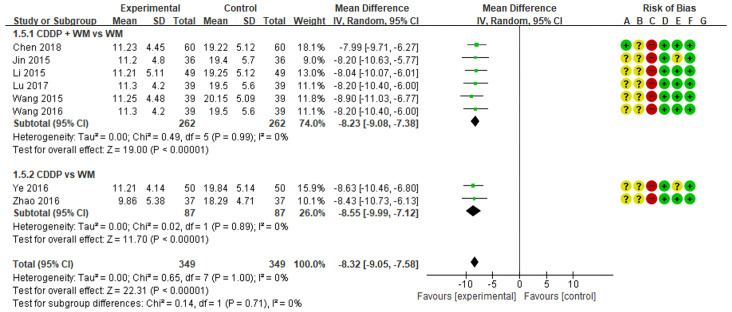
Meta-analysis of homocysteine between treatment and control groups. Green ‘+’ circles = Low risk of Bias; yellow ‘?’ circles = uncertainties about the risk of bias; red ‘−’ circles = high risk of bias. The black rectangle represents the mean difference; The green dot and bar represent the overall summary estimate of the treatment effect, especially, the bar represents the confidence interval for the summary estimate. Reference: Chen 2018 [53]; Jin 2015 [54]; Li 2015 [56]; Lu 2017 [55]; Wang 2015 [58]; Wang 2016 [57]; Ye 2016 [51]; Zhao 2016 [52].

**Figure 7 medicina-59-01730-f007:**
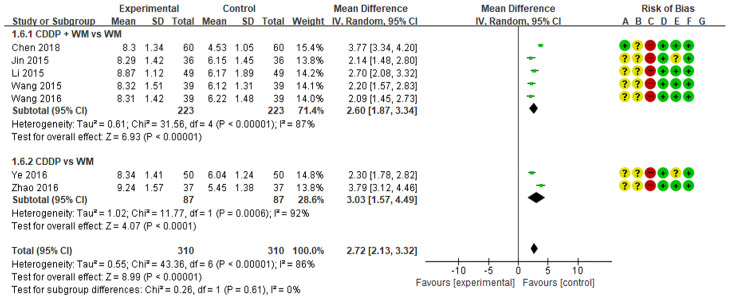
Meta-analysis of plasma adiponectin levels between treatment and control groups. Green ‘+’ circles = Low risk of Bias; yellow ‘?’ circles = uncertainties about the risk of bias; red ‘−’ circles = high risk of bias. The black rectangle represents the mean difference; The green dot and bar represent the overall summary estimate of the treatment effect, especially, the bar represents the confidence interval for the summary estimate. Reference: Chen 2018 [53]; Jin 2015 [54]; Li 2015 [56]; Wang 2015 [58]; Wang 2016 [57]; Ye 2016 [51]; Zhao 2016 [52].

**Figure 8 medicina-59-01730-f008:**
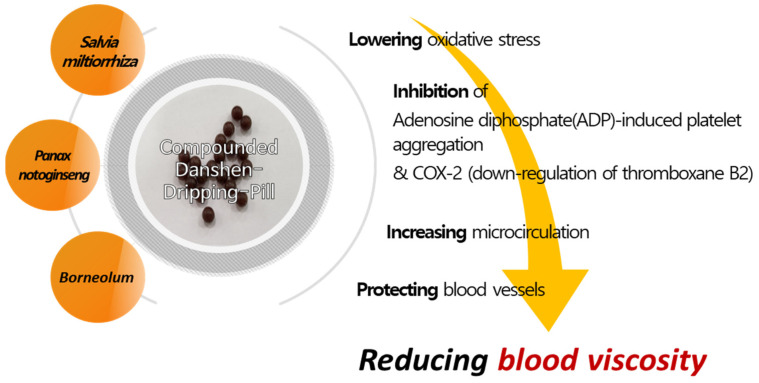
Possible mechanistic pathway of CDDP for reducing blood viscosity.

## Data Availability

The data presented in the study are included in the article and Appendix A. Further inquiries can be directed to the corresponding authors.

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
