# Peer review of "Effectiveness and Safety of Fufang Danshen Dripping Pill (Cardiotonic Pill) on Blood Viscosity and Hemorheological Factors for Cardiovascular Event Prevention in Patients with Type 2 Diabetes Mellitus: Systematic Review and Meta-Analysis"

_medicina, 2023, doi:10.3390/medicina59101730_

Round 1

Reviewer 1 Report

Dear authors,

I have read with interest your paper. I think it is a fairly designed study, yet I have some concerns to address:

- In the Abstract, you state that CDDP is widely used in Asia, yet you have chosen only articles from China, and in line 83 you state that is used in China. You should clarify.

- I think the part in the Introduction where you discuss the classification of Diabetes is not required, as it is well-known general knowledge (Lines 56-58). Type 2 DM could be abbreviated by inserting a phrase in the first part of the Introduction.

- Why didn't you search the Clarivate Web of Science?

- Line 129: You can replace the disease with T2DM.

- The word ”Hemorheologic factors” from the title must be articulated as ”Hemorheological factors”. Furthermore, it exists in the title and in the Introduction, yet it is not even mentioned in the Discussions.

- You did not erase the duplicate Conclusions title from lines 497-499.

- I like the separate chapter 4.5. Strengths and limitations. 

- I feel there is room for improvement in the discussions, and you should pay attention to the English language, in order to make it more easily readable. 

Good luck!

Minor changes are required.

Author Response

1) In the Abstract, you state that CDDP is widely used in Asia, yet you have chosen only articles from China, and in line 83 you state that is used in China. You should clarify.

Answer: Thank you for your comments; we changed abstract ‘widely used in Asia’ to ‘widely used in China’.

2) I think the part in the Introduction where you discuss the classification of Diabetes is not required, as it is well-known general knowledge (Lines 56-58). Type 2 DM could be abbreviated by inserting a phrase in the first part of the Introduction.

Answer: Thank you for your comments; We changed Line 56-59 ‘Diabetes can be classified into type 1, type 2, and gestational diabetes. Type 1 diabetes re-sults from the congenital destruction of pancreatic β-cells that produce insulin, whereas type 2 diabetes (T2DM) is acquired [7,8]. As the prevalence of T2DM increases, so does the risk of cardiovascular disease [9].’ as ‘As the prevalence of T2DM increases, so does the risk of cardiovascular disease [7-9].’  

Furthermore, we add abbreviation of T2DM in the introduction`s first part ‘Diabetes mellitus is a metabolic disease characterized by insulin resistance in the target organs or an absolute or relative insulin deficiency and can be classified into type 1 diabetes mellitus, type 2 diabetes mellitus (T2DM), and gestational diabetes.’

3) Why didn't you search the Clarivate Web of Science?

Answer: Thanks for your kind revision. We did not conduct separate searches for Web of Science, as we anticipated that the results would be similar to those obtained from MEDLINE (because of it include SCIE, SCI, and ESCI journals) and EMBASE. The strength of this study lies in the systematic search conducted in the Chinese language database, CNKI. Furthermore, as reviewer`s kind comment, we will include in the discussion section the limitation that additional studies may be incorporated when searching in databases such as Web of Science and others.

(Line 465-467) “Furthermore, We searched several core databases, including Pubmed, EMBASE, CENTRAL, but we did not review all databases such as Web of Science. Therefore, it is possible that studies not included in our search may exist.”

4) Line 129: You can replace the disease with T2DM.

Answer: Thanks for your comment. We changed Line 129 ‘Eligible participants were patients diagnosed with type 2 diabetes’ as ‘Eligible participants were patients diagnosed with T2DM’

5) The word ”Hemorheologic factors” from the title must be articulated as ”Hemorheological factors”. Furthermore, it exists in the title and in the Introduction, yet it is not even mentioned in the Discussions.

Answer: Thanks for the kind revision. We changed our title as ‘Effectiveness and Safety of Fufang Danshen Dripping Pill (Cardiotonic Pill) on Blood Viscosity and Hemorheological Factors for Cardiovascular Event Prevention in Patients with Type 2 Diabetes Mellitus: Systematic Review and Meta-analysis’

6) You did not erase the duplicate Conclusions title from lines 497-499.

Answer: Thanks for the kind revision. We erase Line 497-499.

7)I like the separate chapter 4.5. Strengths and limitations.

Answer: Thanks for your comment. We separate ‘4.5. Strength and limitations’ as ‘4.5. Strengths’ And ‘4.6. Limitations’

8) I feel there is room for improvement in the discussions, and you should pay attention to the English language, in order to make it more easily readable.

Answer: Thanks for the kind comment. We have made some improvements to the discussion part. Thank you.

Reviewer 2 Report

The purpose of this systematic review is to analyze the effectiveness and safety of CDDP in the blood viscosity (BV) with type 2 diabetes mellitus (T2DM). Actually, the current proposal is interesting and well-written. Therefore, I recommend that the current study be published after minor revisions as follows:

1-   Could the authors discuss obesity as a risk factor for T2DM?

Reference: GATA3 as an immunomodulator in obesity-related metabolic dysfunction associated with fatty liver disease, insulin resistance, and type 2 diabetes. Chem Biol Interact. 2022 Oct 1;366:110141. doi: 10.1016/j.cbi.2022.110141.

2-    Please underline the knowledge gap in the hunt for alternative medication for T2DM. Reference: MD Simulation Studies for Selective Phytochemicals as Potential Inhibitors against Major Biological Targets of Diabetic Nephropathy. Molecules. 2022 Aug 5;27(15):4980. doi: 10.3390/molecules27154980.

3-   Please add a diagrammatic figure to propose the possible mechanistic pathway for these findings

Author Response

Thanks for the kind review. As per the reviewer's suggestions, we have added references to the discussion and introduction sections of the paper. We appreciate your feedback.

1-   Could the authors discuss obesity as a risk factor for T2DM?

Answer: We have added a sentence stating that obesity is one of the risk factors for T2DM, emphasizing the need to consider obesity as a variable in future clinical studies.

2-    Please underline the knowledge gap in the hunt for alternative medication for T2DM.

Answer: We add importance of in silico study. Thanks for your kind revision.

Research on natural-based candidate substances for the treatment of diabetes complications is also conducted in silico, so future in silico-based studies on the constituents of CDDP may aid in understanding their mechanisms [88].

3-   Please add a diagrammatic figure to propose the possible mechanistic pathway for these findings

Answer: As per the reviewer's suggestion, we have added figure 8 to the paper. Thank you.